



# Validating physical and semi-empirical satellite-based irradiance retrievals using high- and low-accuracy radiometric observations in a monsoon-influenced continental climate

Yun Chen[1,2], Dazhi Yang[3], Chunlin Huang[4], Hongrong Shi[5], Adam R. Jensen[6], Xiang'ao Xia[5,7], Yves-Marie Saint-Drenan[8], Christian A. Gueymard[9], Martin János Mayer[10], and Yanbo Shen[1,2]

[1]Public Meteorological Service Centre, China Meteorological Administration, Beijing, China
[2]Key Laboratory of Energy Meteorology, China Meteorological Administration, Beijing, China
[3]School of Electrical Engineering and Automation, Harbin Institute of Technology, Harbin, Heilongjiang, China
[4]Institute of Light Resources and Environmental Sciences, Henan Academy of Sciences, Zhengzhou, Henan, China
[5]Institute of Atmospheric Physics, Chinese Academy of Sciences, Beijing, China
[6]Department of Civil and Mechanical Engineering, Technical University of Denmark, Kgs. Lyngby, Denmark
[7]University of Chinese Academy of Sciences, Beijing, China
[8]MINES ParisTech, PSL Research University, O.I.E. Centre Observation, Impacts, Energy, 06904, Sophia Antipolis, France
[9]Solar Consulting Services, Colebrook, NH, USA
[10]Department of Energy Engineering, Faculty of Mechanical Engineering, Budapest University of Technology and Economics, Műegyetem rkp. 3, H-1111 Budapest, Hungary

**Correspondence:** Dazhi Yang (yangdazhi.nus@gmail.com) and Yanbo Shen (shenyb@cma.gov.cn )

**Abstract.** Are high-accuracy radiometric observations strictly indispensable for the validation of satellite-based irradiance retrievals, or might low-accuracy observations serve as adequate substitutes? Owing to the scarcity of sites with redundant radiometers, such inquiries have seldom been contemplated, much less subjected to systematic examination; rather, it has been customary to employ all accessible observations during validation, frequently with only minimal quality control. In this inves-

tigation, we address this question by validating two distinct sets of satellite-retrieved irradiance—one derived through physical methods, the other through statistical means—against collocated high- and low-accuracy observations. Departing from the majority of validation studies, which rely exclusively upon an array of performance measures, we advocate and implement a rigorous distribution-oriented validation framework, yielding more profound insights and more comprehensive conclusions. Beyond the validation methodology itself, the dataset utilized in this study is noteworthy in its own regard: It incorporates ra-

diometric observations from the newly established and first-ever Baseline Surface Radiation Network (BSRN) station situated within a monsoon-influenced continental climate (specifically, the Dwa Köppen classification), in conjunction with irradiance retrievals from the Fengyun-4B geostationary satellite, which are likewise new to the community. The accumulated evidence strongly suggests that the use of low-accuracy observations as a reference in validating irradiance retrievals may entail significant risks, because the discrepancies they introduce can be of a magnitude comparable to the commonly accepted margins of

error or improvement (approximately several $\mathrm{W\,m^{-2}}$ or a few percent) upon which numerous scientific assertions depend.



## 1 Introduction

Gridded irradiance retrieved from onboard imagers of geostationary satellites constitutes the foundation of numerous endeavors in solar energy meteorology, among which solar resource assessment and forecasting stand out as the most representative (Yang and Kleissl, 2024). Solar resource assessment aims to quantify the long-term (over years or decades) availability and variability of irradiance, thereby enabling the estimation of the energy yield and the evaluation of the economic feasibility and bankability of solar energy projects prior to their deployment. Given that the establishment and maintenance of dense radiometer networks over extensive regions and prolonged durations has never been a practical undertaking, and given that modeled irradiance (such as that derived from numerical weather predictions and reanalysis) remains constrained by its limited accuracy, the use of satellite-retrieved irradiance is the industry-standard practice for such assessments (Yang et al., 2022b). Solar forecasting—across varying short-term scales (from seconds to days)—is indispensable for grid operators, who must reliably formulate generation schedules accounting for the variable and intermittent nature of solar power output. In particular, within the intraday window of 0–4 hours, the advection of (normalized) satellite-retrieved irradiance fields has been consistently demonstrated to be the most reliable strategy (Yang et al., 2022a). In these applications of solar energy meteorology, the assurance of quality in satellite-retrieved irradiance products is of paramount importance—a necessity that has given rise to a substantial body of validation studies (e.g., Elias et al., 2024; Wandji Nyamsi et al., 2023; Qin et al., 2022; Kosmopoulos et al., 2018).

Irradiance retrieval from the top-of-the-atmosphere reflectance and brightness temperature measurements can be categorized into physical and data-driven types, depending on whether the retrieval process incorporates radiative transfer (Huang et al., 2019). Given that the execution of line-by-line radiative transfer calculations for each individual pixel and time stamp is computationally prohibitive, accelerated calculations, such as using a lookup table (e.g., Huttunen et al., 2016; Lefèvre et al., 2013) or parameterization (e.g., Huang et al., 2018; Xie et al., 2016), are invariably adopted in practice. Although physical retrieval methods are firmly grounded in established theory (Liou, 2002), they are not without their difficulties; persistent challenges, such as the parallax effect, that lead to geographic displacement of clouds, remain unresolved. On the other hand, data-driven retrieval techniques, which can be further subdivided into semi-empirical (e.g., Huang et al., 2025; Perez et al., 2002) and machine learning (e.g., Shi et al., 2023; Verbois et al., 2023) methods, offer ease of implementation and can, under favorable conditions, achieve accuracy rivaling that of physical methods. However, such accuracy is contingent upon the quantity and quality of ground-based observations and cannot be assured *a priori*. Indeed, data-driven techniques often leverage large datasets with lower quality due to limited instrument maintenance, and the model potentially learns measurement errors, such as those due to soiling or calibration drift. Documented instances also exist where machine learning models, owing to overfitting, have produced retrievals exhibiting unphysical values or behavior (Yang et al., 2022c). In any case, these inherent limitations of existing retrieval techniques underscore the necessity of formal validation of satellite-derived irradiance.

The quality of satellite-derived irradiance varies as a function of spatial locations, time periods, and atmospheric conditions. Consequently, numerous validation efforts have adopted the practice of partitioning data samples into distinct groups representative of disparate climate classifications, seasonal variations, and sky states. This kind of practice requires extensive data support, where the Baseline Surface Radiation Network (BSRN; Driemel et al., 2018; Ohmura et al., 1998), being the





world's largest research-grade surface radiation monitoring network, has hitherto been an indispensable and reliable data source
(Bright, 2019). BSRN began collecting surface radiation data in 1992 with nine initial stations. Ever since, the station listing
has been dynamic, with some new stations being added and some older ones being closed; the present tally of active stations
stands at 43. One of the initial aims of the network is to achieve coverage across major climate zones (Ohmura et al., 1998).
However, as revealed in a global overview of multi-component solar radiometer monitoring stations (Jensen et al., 2025), the

geographical distribution of climate zones and thus radiometric stations is highly uneven. For that, the establishment of BSRN
in rarer climate zones has proven particularly challenging. For instance, Fig. 1 shows the geographical distribution of the con-
tinental dry winter climates, denoted by the Köppen symbol "Dw." Until recently, its four subzones (Dwa to Dwd) remained
entirely unrepresented within the BSRN framework, constituting a critical factor that precluded systematic investigation under
such meteorological regimes. This impediment has now been partially remedied by the establishment of a new BSRN station

in Qiqihar, China, in October 2023—a development that permits, for the first time, rigorous inquiry into surface radiation
characteristics within continental dry winter climates, including the validation of satellite-retrieved irradiance products.

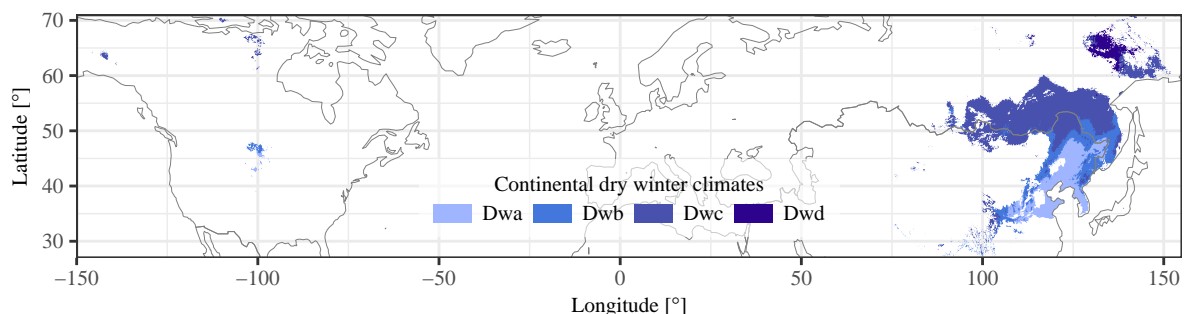

**Figure 1.** The geographical distribution of the continental dry winter climates.

Indeed, ground-based radiometric observations form an indispensable element in the validation of satellite-retrieved irra-
diance. The conventional validation methodology consists of computing aggregate performance measures, such as mean bias
error (MBE), root mean square error (RMSE), or correlation, between retrievals and observations. However, these overall

performance statistics suffer from well-documented limitations, such as interdependence or incompleteness, rendering con-
clusions drawn therefrom inherently questionable (Tian et al., 2016). To give perspective, two distinct sets of retrievals may
yield identical MBE or RMSE values while exhibiting radically divergent temporal characteristics; this phenomenon is known
as underdetermination (Tian et al., 2016) and has been more extensively documented in the forecasting literature (e.g., Yang
and Perez, 2019; Vallance et al., 2017). Therefore, some have recommended using a more rigorous and systematic approach

known as the distribution-oriented validation framework (Murphy and Winkler, 1987), which is receiving increasing accep-
tance in the solar community (Yang and Bright, 2020). The key philosophy of the distribution-oriented validation framework
is to scrutinize the joint, marginal, and conditional distributions of prediction and observation, which allows visual and quan-
titative assessments of different aspects of quality, such as bias, association, calibration, refinement, or discrimination, through




decomposition of the RMSE (Yang et al., 2020). These aspects of quality possess clearly defined statistical complementar-
ity, thereby offering more substantive insight than traditional measure-oriented validation, which often proves both subjective
and inconclusive. To that end, this work employs the distribution-oriented validation framework, thereby furnishing a more
comprehensive and analytically sound assessment of the irradiance products under examination.

Beyond questions of validation methodology lies a more fundamental concern regarding the quality of ground-based ra-
diometric observations themselves. Since these observations serve as the ground truth, their uncertainties and biases, which
are typically unaccounted for, inevitably undermine the confidence of any derived conclusions. Although the manufacturer-
reported daily uncertainty of secondary-standard (i.e., Class A) radiometers is below 2%, such singular numerical representa-
tions cannot possibly represent the true uncertainties encountered under actual field conditions. It has been reported that the
difference among regularly calibrated thermopile radiometers can reach up to $\pm 17\%$ at high zenith angles and under cloudy
skies (Habte et al., 2016). Moreover, the impracticality of deploying redundant radiometers for repeated measurements com-
pels reliance upon rigorous quality control (QC) procedures as the sole means of ensuring the validity of ground truth. Even so,
most operational radiometric sites are limited to measuring the global irradiance component, and not the individual diffuse and
beam components. This limitation renders certain QC tests inapplicable, most notably the three-component closure test, which
verifies the agreement between measured global irradiance and the sum of its diffuse and beam constituents. These considera-
tions lead us to a critical epistemological question: To what extent can observed validation deviations be properly attributed to
limitations in the ground-based observations themselves, rather than to deficiencies in the satellite retrievals? This distinction
is not merely technical but fundamental to the proper interpretation of validation results (see Section 10.4 of Sengupta et al.,
2024).

Guided by the above preliminaries, the present investigation establishes three principal objectives: (1) conduct a systematic
validation of physical and semi-empirical satellite irradiance retrievals under the monsoon-influenced dry-winter hot-summer
continental climate (i.e., the Dwa Köppen climate), which is a climatic regime remarkably underrepresented in existing studies;
(2) demonstrate the methodological superiority of distribution-oriented validation compared to conventional measure-oriented
approaches; and (3) quantitatively assess the propagation of observational uncertainties from ground-based radiometers into
validation outcomes. The organization of this paper proceeds as follows: Section 2 provides a comprehensive exposition of the
datasets, their quality control procedure, and the formal validation methodology. Section 3 depicts the validation outcomes and
discusses the implications of the quality of ground-truth on gridded irradiance validation. Section 4 synthesizes the principal
findings and their broader implications.

## 2 Data and method

### 2.1 Data description

A total of four datasets are involved herein, among which two consist of ground-based radiometric observations, and the
other two are satellite-retrieved irradiance. The two sets of ground-based observations are collocated. One set of ground-based
observations comes from the BSRN Qiqihar (QIQ) station, which employs secondary-standard radiometers and represents



high-standard radiometric practices. In contrast, the other set of ground-based observations comes from an operational station of the China Meteorological Administration (CMA), which uses a pyranometer from a Chinese manufacturer that is potentially associated with higher uncertainties than the secondary-standard ones. Notation-wise, we denote the high-accuracy set of observations as $y_H$, and the low-accuracy set of observations as $y_L$.

The two sets of satellite-retrieved irradiance are both derived from images captured by the Advanced Geostationary Radiation Imager (AGRI) on board the Fengyun-4B (FY-4B) satellite, which is the second of the latest-generation Chinese geostationary meteorological satellite series. To exemplify the physical retrieval algorithms, the official FY-4B irradiance product of the National Satellite Meteorological Center (NSMC) of CMA is considered. As for the statistical (or semi-empirical) retrieval, the product of Huang et al. (2023), which is based on a modified version of the Heliosat-2 method of Rigollier et al. (2004), is used without loss of generality. Similarly to the case of ground-based observations, the two sets of satellite-retrieved irradiance are denoted as $x_P$ and $x_S$, respectively, with the subscripts denoting "physical" and "statistical." In short, this work seeks to validate $x_P$ and $x_S$ using $y_H$ and $y_L$, separately. The period of validation is one year from April 2024 to March 2025.

### 2.1.1 High-accuracy observations from the BSRN QIQ station

The BSRN QIQ station (47.7957° N, 124.4852° E, 170 m) is located in the monsoon-influenced dry-winter hot-summer continental climate, which has a symbol of "Dwa" under the Köppen climate classification. Given the geographical distribution of the Dwa climate, which is mostly in northeastern Asia, the surface radiation characteristics based on high-quality ground-based radiometers have rarely, if ever, been formally reported, thereby making this report unique and valuable.

Following the BSRN requirements, QIQ measures all three downward surface shortwave radiation components, as well as the downward longwave radiation, with a sampling rate of 1 s. For the global horizontal irradiance (GHI) and diffuse horizontal irradiance (DHI), CMP22 pyranometers from Kipp & Zonen are used, whereas a CHP1 pyrheliometer mounted on a SOLYS2 sun tracker is used to measure the beam normal irradiance (BNI). All radiometers are calibrated and regularly maintained. For more technical details of the QIQ station, the reader is referred to the recent work by Liu et al. (2025). It should be highlighted that the CMP22 pyranometers are ventilated but not heated, due to the high thermal offset observed during the winter of 2023; the heaters have been switched off ever since.

### 2.1.2 Low-accuracy observations from the CMA operational station

Compared to basic meteorological variables, such as temperature or precipitation, solar irradiance does not have the same degree of impact on daily human activities. Consequently, radiometers have hitherto been regarded as an additional feature in weather stations. For instance, CMA maintains more than 50,000 manned and unmanned weather stations, but only about 100 of them have radiometers installed (Yang et al., 2022b). Among these CMA operational radiometric stations, only a handful measure all three shortwave components, whereas most only measure GHI, including the Fuyu County Meteorological Bureau, within which the QIQ station is located. This GHI dataset, therefore, serves as an alternative version of ground truth for validating gridded irradiance retrievals.



The radiometer employed by the Fuyu Bureau is a DFN1 thermopile pyranometer by Huatron Environment, a Chinese brand. DFN1 is a first-class (i.e., Class B) pyranometer that has acquired the CMA's special technical equipment license for meteorological observation. According to the manufacturer, DFN1 has a spectral range of 300–3000 nm (200–3600 nm for CMP22), a response time of $<20$ s ($<5$ s for CMP22), and a temperature response of $\leq \pm 2\%$ ($\leq \pm 0.5\%$ for CMP22). The data is sampled at 2 s but only logged as 1-min averages. The DFN1 at the Fuyu Bureau is also equipped with an SRC228 ventilation unit. Through the CMA standard data and communication protocol, the 1-min data is transmitted to the central server in near real-time.

### 2.1.3 Physical irradiance retrievals from NSMC

The FY-4B satellite was successfully launched on June 3, 2021, and was located at $123.5°$ E on June 10, 2021, relocated at $133°$ E from April 11, 2022, to January 31, 2024, and positioned at its final position of $105°$ E on March 5, 2024. The NSMC official FY-4B irradiance product is developed by the Algorithm Working Group and consists of all three surface shortwave components. The retrieval algorithm is of a physical type, which, as mentioned in the introduction, is based on radiation transfer, accounting for all important radiation transfer physical processes in the shortwave range, including multiple scattering, absorption, and thermal radiation, to form a look-up table. More specifically, the observations from channels 1 to 6 of the FY-4B AGRI and the FY-4B L2 snow cover product are used to obtain the instantaneous state variable information of the atmosphere and the surface, mainly including atmospheric extinction parameters and surface reflectance. After determining the instantaneous state of the atmosphere and the surface, irradiance is estimated according to the pre-established lookup table, taking into account the sun–surface–satellite geometry.

FY-4B has 15 channels with a native resolution ranging from 0.5–4 km at nadir, depending on the channel. Since the retrieval algorithm uses multiple channels, the spatial resolution of the final product follows that of the lowest-resolution channel, which is 4 km. Temporally, AGRI completes a full-disc scan over each 15-min interval. The exact instant of AGRI passing the pixel corresponding to the QIQ station is not considered in this work. Instead, the time stamps follow the scan intervals. This slight asynchrony is unlikely to cause major offsets during validation, especially after hourly averages are formed.

### 2.1.4 Statistical irradiance retrievals using the modified Heliosat-2 method

Heliosat-2 is a well-known statistical irradiance retrieval technique that is primarily based on three components: (1) a clear-sky model, (2) a method to estimate cloud index from satellite reflectance observations, and (3) a mapping function from cloud index ($\nu$) to clear-sky index ($\kappa$). Once the $\kappa$ is obtained, GHI can be reconstructed by multiplying $\kappa$ with the clear-sky GHI. The reader is referred to Appendix A for an extended explanation of the algorithm. In the work of Huang et al. (2023), the clear-sky GHI is estimated via the REST2 model, the cloud index is estimated from the Fengyun-4A (FY-4A) level-1 reflectance of the visible channel (0.65 μm), and the coefficients of the $\nu$-to-$\kappa$ mapping are determined based on 38 stations operated by the Chinese Ecosystem Research Network. Since only the visible channel observations are involved, the spatial resolution of the product is 0.5 km, which can reveal more granular spatial features without sacrificing much accuracy compared to a benchmark product—the Himawari-8 product developed by the Japan Aerospace Exploration Agency (Huang et al., 2023). On




the other hand, the cloud shadow and edge reflectances are more pronounced for high-resolution retrievals, which can disturb the calculation of the cloud index.

In this work, the same retrieval technique is used to acquire GHI on a grid cell collocated with the QIQ station, with the only difference being the raw data source, which is FY-4B instead of FY-4A. It should be noted that the FY-4A is an experimental satellite with known fluctuations in radiometric performance and an irregular observation schedule (Zhong et al., 2021). In contrast, the FY-4B technology is more mature than FY-4A, and the performance is comparable to the GOES-R and Himawari-8/-9 satellites; also, it makes a full-disc scan every 15 min. The statistically retrieved irradiance, therefore, follows the observation schedule of FY-4B, which has a regular 15-min interval.

## 2.2 Data quality control and temporal alignment

To ensure the soundness of the outcome drawn from the validation exercise, the ground-based observations must first undergo quality control. Additionally, considering the differences in temporal resolutions of various datasets, it is vital to aggregate and align them temporally. These two aspects are described in this section.

### 2.2.1 Quality control routine for irradiance

The optimal QC procedure ought to provide a sufficient balance between the number of rejected inlier data points and the number of retained anomalous data points. In this work, the QC checks used by the International Energy Agency (IEA) PVPS Task 16 team are considered (Forstinger et al., 2021), which is based on the BSRN recommended routine Long and Shi (2008) with notable additions. Notation-wise, several well-known normalized indexes (or $k$ indexes) are to be first defined, they are the clearness index $k_t = G_h/E_0$, the beam transmittance $k_b = B_n/E_{0n}$, the clear-sky index $\kappa = G_h/G_{hc}$, and the diffuse fraction $k = D_h/G_h$, where $G_h$, $D_h$, and $B_n$ denote GHI, DHI, and BNI, respectively; $G_{hc}$ is the clear-sky GHI; and $E_0 = E_{0n}\cos Z$ is the extraterrestial GHI, with $Z$ being the zenith angle.

The $k$ indexes are normalized versions of various irradiance components; they are also very commonly used in radiation modeling, when the yearly and diurnal cycles of irradiance interfere with model fitting (Yang and Kleissl, 2024). Based on the $k$ indexes, several QC tests can be written:

- $k_b < k_t$, for $G_h > 50\,\mathrm{W\,m^{-2}}$, $k_t > 0$, and $k_b > 0$;

- $k_b < (1100\,\mathrm{W\,m^{-2}} + 0.03 \times \text{altitude in m.a.s.l.})/E_{0n}$, for $G_h > 50\,\mathrm{W\,m^{-2}}$ and $k_b > 0$;

- $k_t < 1.35$, for $G_h > 50\,\mathrm{W\,m^{-2}}$;

- $k < 1.05$, for $Z < 75°$ and $G_h > 50\,\mathrm{W\,m^{-2}}$;

- $k < 1.10$, for $Z \geq 75°$ and $G_h > 50\,\mathrm{W\,m^{-2}}$;

- $k < 0.96$, for $k_t > 0.6$, $Z < 85°$, and $G_h > 50\,\mathrm{W\,m^{-2}}$.





When the conditions specified by these tests are *not* met, the corresponding data points are flagged as "anomalous."

Besides the $k$ index tests, the extremely rare limits (ERL) test and the three-component closure test are also part of the QC routine. They are:

- $-2\,\mathrm{W\,m^{-2}} \leq G_h \leq 1.2 E_{0n} \cos^{1.2} Z + 50\,\mathrm{W\,m^{-2}}$,

- $-2\,\mathrm{W\,m^{-2}} \leq D_h \leq 0.75 E_{0n} \cos^{1.2} Z + 30\,\mathrm{W\,m^{-2}}$,

- $-2\,\mathrm{W\,m^{-2}} \leq B_n \leq 0.95 E_{0n} \cos^{0.2} Z + 10\,\mathrm{W\,m^{-2}}$;

and

- $\mathrm{abs}(\delta) \leq 8\%$, for $Z < 75°$ and $G_h > 50\,\mathrm{W\,m^{-2}}$,

- $\mathrm{abs}(\delta) \leq 15\%$, for $93° > Z > 75°$ and $G_h > 50\,\mathrm{W\,m^{-2}}$;

where $\delta = G_h/(B_n \cos Z + D_n) - 1$ is the decimal difference of the closure relationship. Last but not least, the tracker-off test is given by:

- $(G_{hc} - G_h)/(G_{hc} + G_h) < 0.2$, for $Z < 85°$;

- $(B_{nc} - B_n)/(B_{nc} + B_n) > 0.95$, for $Z < 85°$.

It should be noted that although $G_{hc}$ and $B_{nc}$ can be estimated via a clear-sky model, they are defined as $G_{hc} = 0.8 E_0$ and $B_{nc} = (G_{hc} - D_{hc})/\cos Z$ herein, with $D_{hc} = 0.165 G_{hc}$.

The above QC routine requires all three irradiance components to execute, and thus can only be applied to the QIQ observations. As for the CMA observations, only the first line of the ERL test is applicable. All tests are performed on 1-min data, and the outcome is displayed in Fig. 2, which is known as a multiplot, as named by Forstinger et al. (2021), to whom the reader is referred for detailed interpretation. (There are several calculated quantities with subscript "calc" in this plot, which all result from the closure equation.) In short, the 1-min QIQ observations are of good quality with very few missing and rejected data points. Most of the rejected data points are due to (1) frost deposition on the glass cover of the pyrheliometer, (2) elevated thermal offsets in pyranometers, and (3) horizon shading by nearby buildings during winter sunsets. Although regular maintenance is conducted by the on-site staff, it is still difficult to entirely avoid data issues, again highlighting the importance of QC.

### 2.2.2 Data aggregation and temporal alignment

One year of 1-min observations corresponds to 525,600 samples, among which 239,879 are daytime samples—considering the higher directional uncertainty of pyranometers at low-sun conditions, daytime samples are defined as those with zenith angles $\leq 85°$. The daytime samples from QIQ are passed through the above-mentioned QC routine, and 1510 (or 0.64%) data points are flagged and thus rejected before aggregation. The non-flagged data are aggregated to a 15-min data interval using right-labeled timestamps; i.e., the time stamps correspond to the end of the aggregation interval. To ensure that each aggregated





**Figure 2.** A multiplot depicting the quality of QIQ observations.

value can sufficiently represent the average condition of the corresponding intervals, only intervals with more than seven 1-min data points are considered valid. After aggregation, a total of 15,576 valid 15-min data points remain.

After processing the QIQ data (i.e., $y_{\mathrm{H}}$), the 1-min CMA observations (i.e., $y_{\mathrm{L}}$) are aggregated in the same fashion and appended to the QIQ data frame, alongside the two sets of 15-min-resolution retrievals (i.e., $x_{\mathrm{P}}$ and $x_{\mathrm{S}}$). It is noted that both the





CMA observations and the retrievals contain missing instances. A row-wise filter is thus applied to the data frame to remove
incomplete timestamps, resulting in 14,595 final samples. Stated differently, all subsequent validation exercises are based on
this $14,595 \times 4$ data frame, with the four columns being $y_\mathrm{H}$, $y_\mathrm{L}$, $x_\mathrm{P}$, and $x_\mathrm{S}$. Figure 3(a) shows the scatterplot between $y_\mathrm{H}$
and $y_\mathrm{L}$, and the high- and low-accuracy observations have an exceptional agreement, with a Pearson correlation of 0.997 and
a mean difference of 3 W m$^{-2}$. In contrast, the scatterplot between $x_\mathrm{P}$ and $x_\mathrm{S}$, as depicted in Fig. 3(b), shows substantial
discrepancies and noticeable nonlinearity, with a correlation of 0.926 and a mean difference of 40 W m$^{-2}$, suggesting that one
of these products has major technical issues that need to be further investigated.

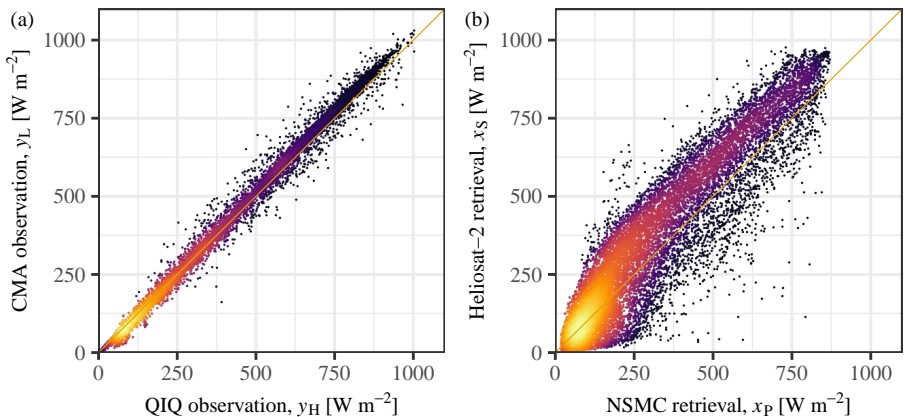

**Figure 3.** Scatterplots of (a) the low-accuracy CMA observations ($y_\mathrm{L}$) against the high-accuracy QIQ observations ($y_\mathrm{H}$) and (b) the Heliosat-2
statistical retrieval ($x_\mathrm{S}$) against the NSMC's physical retrieval ($x_\mathrm{P}$). Brighter colors signify higher sample density.

### 2.3 An overview of the validation methodology

Validation of gridded products can be categorized into measure- and distribution-oriented approaches. Whereas the former
employs a suite of performance measures and statistics to gauge the product quality, the latter examines the joint (or equiva-
lently, the marginal and conditional) distributions of retrieval and observation. However, these two approaches are not mutually
exclusive, as many performance measures can be written in terms of distributions, see Eqs. (1) and (2) below. Therefore, both
approaches are introduced next and their outcomes reported in Section 3.

#### 2.3.1 Measure-oriented validation

A majority of existing works on gridded irradiance validation use the measure-oriented approach, among which the MBE and
RMSE are the most popular choices of performance measures (e.g., Yang and Bright, 2020; Sengupta et al., 2018). MBE is
often reported because ensuring a low bias has the utmost importance in solar resource assessment, but high accuracy is not
as critical (Perez et al., 2013). On the other hand, RMSE is a measure of accuracy that penalizes larger errors, which are





particularly undesirable during solar forecasting (Yang et al., 2020); it is thus by far the most popular measure (Blaga et al., 2019). Without loss of generality, these two performance measures are used in this work.

Denoting the random variable representing the gridded retrieval as $X$, and that representing the observation as $Y$, then MBE
and RMSE can be written as:

$$\text{MBE}(X,Y) = \mathbb{E}\left(X - Y\right) = \iint (x-y)f(x,y)dxdy, \tag{1}$$

$$\text{RMSE}(X,Y) = \sqrt{\mathbb{E}\left(X - Y\right)^2} = \left[\iint (x-y)^2 f(x,y)dxdy\right]^{\frac{1}{2}}, \tag{2}$$

where $\mathbb{E}$ is the expecation operator, which can be evaluated with $n$ samples, and $f(x,y)$ is the joint probability density function
(PDF) of $X$ and $Y$.

### 2.3.2 Distribution-oriented validation

Distribution-oriented validation allows both visual and quantitative assessments. It is highlighted that, besides the order of data
samples, the joint distribution of retrieval and observation contains all information relevant to validation. According to Bayes'
theorem, the joint PDF of retrieval $X$ and observation $Y$ may be decomposed in two ways:

$$f(x,y) = f(y|x)f(x), \tag{3}$$

$$f(x,y) = f(x|y)f(y), \tag{4}$$

where $f(y|x)$ and $f(x|y)$ are the conditional PDFs of $Y$ given $X$ and $X$ given $Y$, respectively; $f(x)$ and $f(y)$ are marginal
PDFs. Whereas Eq. (3) is known as the calibration–refinement factorization, Eq. (4) is known as the likelihood–base rate
factorization.

The above naming convention reveals how each conditional or marginal distribution is linked to certain characteristics and
properties of the validation samples. For instance, $f(y|x)$ signifies the *calibration* of the retrievals, since it describes whether
or not the observations can reliably materialize according to a particular retrieved value, say $x_0$—the word "reliable," in
this context, means that the expectation of the materialized observations agrees with the retrieved value, or mathematically,
$\mathbb{E}(Y|X = x_0) = x_0$. Next, $f(x)$ is the probability density of retrieval, which describes how *refined* the retrievals are—if the
same value results each time, the retrievals are not at all refined. As for the $f(x|y)$ term, it narrates the *likelihood* of different
values of retrievals would have been issued before a particular observation value, say $y_0$, was materialized; analogous to the
calibration term, one seeks the property $\mathbb{E}(X|Y = y_0) = y_0$ when assessing the likelihood. Last, the probability density of
observation, that is, $f(y)$, is referred to as the *base rate* in meteorology, and is a characteristic of the atmospheric process itself.

The visual assessment of the calibration–refinement and likelihood–base rate factorization can be performed by plotting
out the densities, e.g., one may overlay $f(x)$ with $f(y)$ and observe the discrepancy between them. This type of analysis is
intuitive. For quantitative assessments, the strategy is to decompose the mean square error (MSE) into several terms, each





holding a statistical interpretation. Specifically, MSE has the following well-known decompositions:

$$\text{MSE}(X,Y) = \overbrace{\mathbb{V}(X) + \mathbb{V}(Y)}^{\text{marginal dist.}} - \overbrace{2\,\text{Cov}(X,Y)}^{\text{association}} + \overbrace{[\mathbb{E}(X) - \mathbb{E}(Y)]^2}^{\text{Unconditional bias}}, \tag{5}$$

$$\text{MSE}(X,Y) = \mathbb{V}(Y) + \overbrace{\mathbb{E}_X[X - \mathbb{E}(Y|X)]^2}^{\text{calibration}} - \overbrace{\mathbb{E}_X[\mathbb{E}(Y|X) - \mathbb{E}(Y)]^2}^{\text{resolution}}, \tag{6}$$

$$\text{MSE}(X,Y) = \mathbb{V}(X) + \overbrace{\mathbb{E}_Y[Y - \mathbb{E}(X|Y)]^2}^{\text{type-2 cond. bias}} - \overbrace{\mathbb{E}_Y[\mathbb{E}(X|Y) - \mathbb{E}(X)]^2}^{\text{discrimination}}. \tag{7}$$

Equation (5) is the bias–variance decomposition of MSE, which gives rise to two variance terms, which are summaries of the variability in $f(x)$ and $f(y)$, a covariance term, which explains the linear correlation (i.e., association) between $X$ and $Y$, and a squared unconditional bias term, which quantifies the bias of $X$ with respect to $Y$. Equation (6) is closely tied to the calibration–refinement factorization. The second term of Eq. (6) computes the expected value of calibration—recall that a set of retrievals is said to be calibrated if $\mathbb{E}(Y|X = x) = x$ for all $x$. As for the third term, it calculates the mean

difference between the conditional and unconditional expectations of observation. When a set of retrievals possesses resolving power (i.e., resolution), the difference $\mathbb{E}(Y|X)$ and $\mathbb{E}(Y)$ should be significant. In other words, one expects the observations to materialize differently after different retrieved values are issued. Equation (7) is related to the likelihood–base rate factorization. Its second term is the mean square difference between $Y$ and $\mathbb{E}(X|Y)$, which should be minimized. Suppose for a particular observation $y_0$, the retrievals behave as $\mathbb{E}(X|y = y_0) = y_0 - b$, then we say that the retrieval technique is biased toward an

additive constant $b$ at $y_0$. The third term of Eq. (7) gauges the ability of the retrieval technique in discriminating the diverse atmospheric conditions—if the atmospheric condition as specified by $Y$ has no relevance on how $X$ is generated, the retrieval technique has no discrimination.

## 3    Results and discussion

This section comprises two distinct analytical components. The first component examines all-sample validation, presenting

comprehensive results from both the measure-oriented and distribution-oriented validation approaches in Sections 3.1 and 3.2, respectively. Its primary objective is to resolve the fundamental question underlying this investigation: To what measurable extent does the selection between low- and high-accuracy radiometric observations influence validation outcomes? The second component addresses three specific analytical considerations pertaining to the satellite-retrieved irradiance products under examination: (1) the relationship between temporal averaging intervals and validation metrics, (2) the implications of simple bias

correction on validation results, and (3) identifiable algorithmic limitations that demonstrably compromise retrieval quality; these discussions are allocated to Sections 3.3–3.5.

### 3.1    Overall results of measure-oriented validation

Table 1 shows the overall bias and accuracy of the two satellite-retrieved products at the validation location. Both MBE and RMSE are presented in the same unit as GHI, which is $\text{W m}^{-2}$. Their relative counterparts, i.e., normalized MBE (nMBE) and





normalized RMSE (nRMSE), prefixed with "n" and noted in the parentheses, are normalized with respect to the mean instead of the maximum. Several interesting observations can already be made from Table 1.

**Table 1.** Measure-oriented validation results of $x_P$ and $x_S$, with respect to $y_H$ and $y_L$. Both MBE and RMSE are in $\mathrm{W\,m^{-2}}$, with the normalized metrics noted in parentheses.

| | MBE (nMBE) | | RMSE (nRMSE) | |
| --- | --- | --- | --- | --- |
| | $x_P$ | $x_S$ | $x_P$ | $x_S$ |
| $y_H$ | $-58$ $(-15.6\%)$ | $-18$ $(-4.9\%)$ | $107$ $(28.9\%)$ | $92$ $(25.0\%)$ |
| $y_L$ | $-61$ $(-16.3\%)$ | $-21$ $(-5.7\%)$ | $114$ $(30.4\%)$ | $93$ $(24.9\%)$ |

First, although all cases show negative MBEs, which suggests that both products underestimate GHI, the MBEs referenced to the high-accuracy ground-based observations are smaller than those referenced to the low-accuracy observations. This reveals that bias in observations propagates into the validation results. However, such an effect is directional; that is, only if the bias of the observations and retrievals is of different signs will the apparent bias be larger than the true bias. Second, the differences in bias do not fully propagate into accuracy; the nMBE of $x_S$ with respect to $y_H$ and $y_L$ are $-18$ $\mathrm{W\,m^{-2}}$ and $-21$ $\mathrm{W\,m^{-2}}$, respectively, but the difference between the RMSEs is smaller. This result can be explained through the bias–variance decomposition of MSE, in which bias only contributes partially to the overall accuracy. Third, the RMSEs of $x_S$ evaluated against $y_H$ and $y_L$ are rather indistinguishable, which highlights the main deficiency of measure-oriented validation— the results are often ambiguous and therefore misleading. More specifically, if RMSE is used as the sole performance measure, conclusions such as "$y_H$ and $y_L$ do not make a difference in validating $x_S$" may be yielded.

In any case, the present measure-oriented validation suggests the superiority of the statistically retrieved GHI over the physically retrieved ones. This is not surprising, for the quality of the physical irradiance retrieval algorithms depends highly on the quality of those atmospheric state inputs, such as aerosols or water vapor. However, the NSMC has yet to establish a mature retrieval system for FY-4B (Xia et al., 2025), and directly inputting the imprecise atmospheric states to the retrieval algorithm can lead to unphysical or unrealistic irradiance values, which demand further investigation. However, measure-oriented validation offers only an overview of the product quality but lacks diagnostic ability at large. To that end, distribution-oriented validation ought to be employed if one aims to analyze and improve product quality.

### 3.2 Overall results of distribution-oriented validation

As previewed in Section 2.3, distribution-oriented validation allows both visual and quantitative assessments of the joint distribution of retrieval $(X)$ and observation $(Y)$, which are both considered in the present context as random variables. Since the joint distribution $f(x, y)$ can be factorized into $f(x)$, $f(y|x)$, $f(y)$, and $f(x|y)$ via Bayes' theorem, visual assessment is facilitated by plotting out the joint, marginal, and conditional densities. On the other hand, quantitative assessment under the



distribution-oriented validation mainly revolves around the MSE decompositions, e.g., Eqs. (5)–(7), from which a comprehen-
335 sive collection of aspects of retrieval product quality can be gauged.

### 3.2.1 Visual validation

Given two sets of irradiance retrievals, namely, $x_P$ and $x_S$, alongside two sets of ground-based observations, namely, $y_H$ and
$y_L$, four groups of retrieval–observation pairs can be formed; they are, $(x_P, y_H)$, $(x_S, y_H)$, $(x_P, y_L)$, and $(x_S, y_L)$. Each of these
combinations results in a set of joint, marginal, and conditional densities, which are depicted in Figs. 4–6.
The joint PDFs in Fig. 4 are represented through contour lines superimposed upon $x$–$y$ scatter plots, in which brighter colors
correspond to higher point density. The physical retrieval product manifests its most striking characteristic through systematic
irradiance underestimation, producing a distinctly nonlinear relationship with both the high- and low-accuracy observations. In
an ideal case, scatter points should cluster tightly about the identity line; however, neither the $(x_P, y_H)$ and $(x_P, y_L)$ distributions
exhibit this desired behavior. The underestimation of $x_P$ is rather systematic, as it occurs over almost the entire irradiance
range from 100–1000 $\mathrm{W\,m^{-2}}$, strongly suggesting an inherent algorithmic deficiency. Under normal circumstances, clear-sky
conditions should yield highly accurate retrievals, resulting in dense point clusters along the identity line, as exemplified by the
scatters in Figs. 4 (b) and (d). The observed deviation from this expected pattern in Figs. 4 (a) and (c) leads us to hypothesize
that the physical algorithm employed by NSMC has poor clear-sky irradiance retrieval ability, possibly due to the low and
inconsistent quality of aerosol and water vapor, which serve as essential inputs for clear-sky irradiance computation. In fact,
improving individual stages of the physical irradiance retrieval process has been identified as the foremost scientific challenge
of the current FY-4B products (Xia et al., 2025).

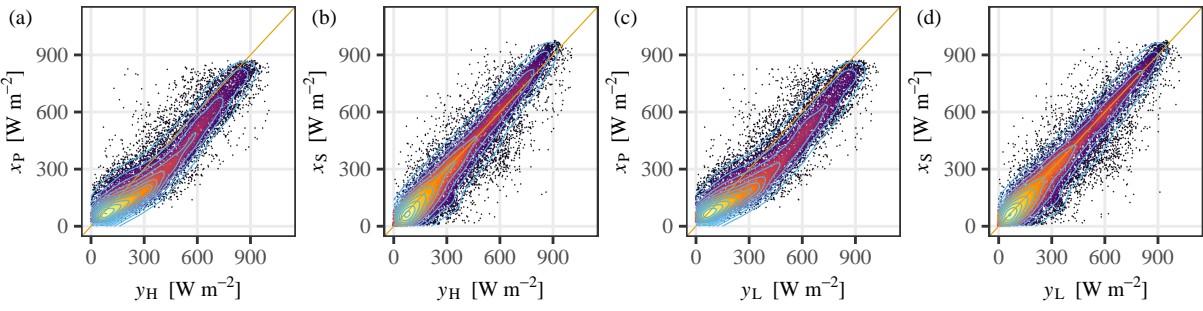

**Figure 4.** The joint PDFs of retrievals and observations. Subplots (a), (b), (c), and (d) correspond to the different combinations of $x_P$, $x_S$, $y_H$, and $y_L$. The contours show the 2D kernel densities.

Figure 5 shows the marginal densities of retrieval and observation, under the four combinations. The number in the top-right
corner of each subplot indicates the Wasserstein distance between two PDFs, which gauges their similarity; the smaller the
better. The disagreements between the physical retrievals with respect to both high- and low-accuracy observations are larger
than those between the statistical retrievals and observations, which is consistent with the finding from Fig. 4. It can be seen




that the PDF of $x_P$ is higher in the low-irradiance range but lower in the high-irradiance range than the PDFs of the two sets of observations, again indicating underestimation. In contrast, the alignment between the PDFs of $x_S$ and observations is much closer. But interestingly, in the case of $x_S$, the validation against $y_L$ returns a lower Wasserstein distance than $y_H$, which signifies the potential danger of an overconfident validation when low-accuracy observations are used as reference.

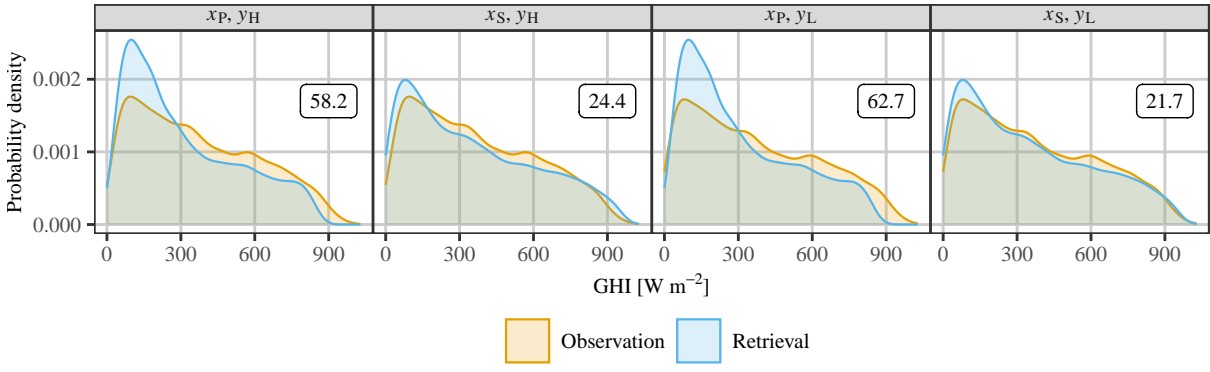

**Figure 5.** The marginal PDFs of retrievals ($x_P$ or $x_S$) and observations ($y_H$ or $y_L$). The number in the top-right corner of each subplot is the Wasserstein distance between the corresponding marginal densities.

Finally, we examine the conditional PDFs presented in Fig. 6, where the upper row displays $f(y|x)$ and the lower row $f(x \mid y)$. The color gradations within these conditional distributions correspond to quantile values, whereas the black dots denote conditional means. It is a fundamental tenet of proper calibration that $\mathbb{E}(Y|X=x)=x$ must hold universally across all $x$. This calibration condition permits direct evaluation of both $x_P$ and $x_S$ through inspection of Fig. 6 (a), where each panel exhibits numerous distinct conditional distributions. The sequence of black dots represents $\mathbb{E}(Y|X=0), \mathbb{E}(Y|X=100), \ldots, \mathbb{E}(Y|X=

1000)$. Consequently, the proximity of these markers to the identity line provides a visual aid for assessing the calibration quality, i.e., the closer to the identity line the black dots are, the smaller the average deviation of $\mathbb{E}(Y|X=x)$ from $x$ is, which in turn implies better calibration. Clearly then, $x_P$ is less calibrated than $x_S$. In an analogous fashion, the quantity $\mathbb{E}(X|Y=y)=y$ can be assessed through Fig. 6 (b), in which the black dots correspond to $\mathbb{E}(X|Y=0), \mathbb{E}(X|Y=100), \ldots, \mathbb{E}(X|Y=1000)$. It is evident that the conditional biases, i.e., $\mathbb{E}(X|Y=y)-y$, for $y=0, 100, \ldots, 1000$, are larger for $x_P$ than for $x_S$, confirming

the superiority of the latter. One should note that the deviations $\mathbb{E}(Y|X=x)-x$ and $\mathbb{E}(X|Y=y)-y$ are directional, therefore, during quantitative evaluation, taking the absolute or squaring the term before averaging is necessary, as shown next.

### 3.2.2    Quantitative validation

Distribution-oriented validation also allows for quantitative assessment of various aspects of product quality. A total of six aspects of product quality, which are based on Eqs. (5)–(7), are considered herein. Given a set of validation samples, with

$X$ denoting the retrieval and $Y$ the observation, the six aspects of quality are association, unconditional bias, calibration, resolution, type-2 conditional bias, and discrimination. They can be quantified through $\text{Cov}(X,Y)$, $[\mathbb{E}(X)-\mathbb{E}(Y)]^2$, $\mathbb{E}_X[X-$





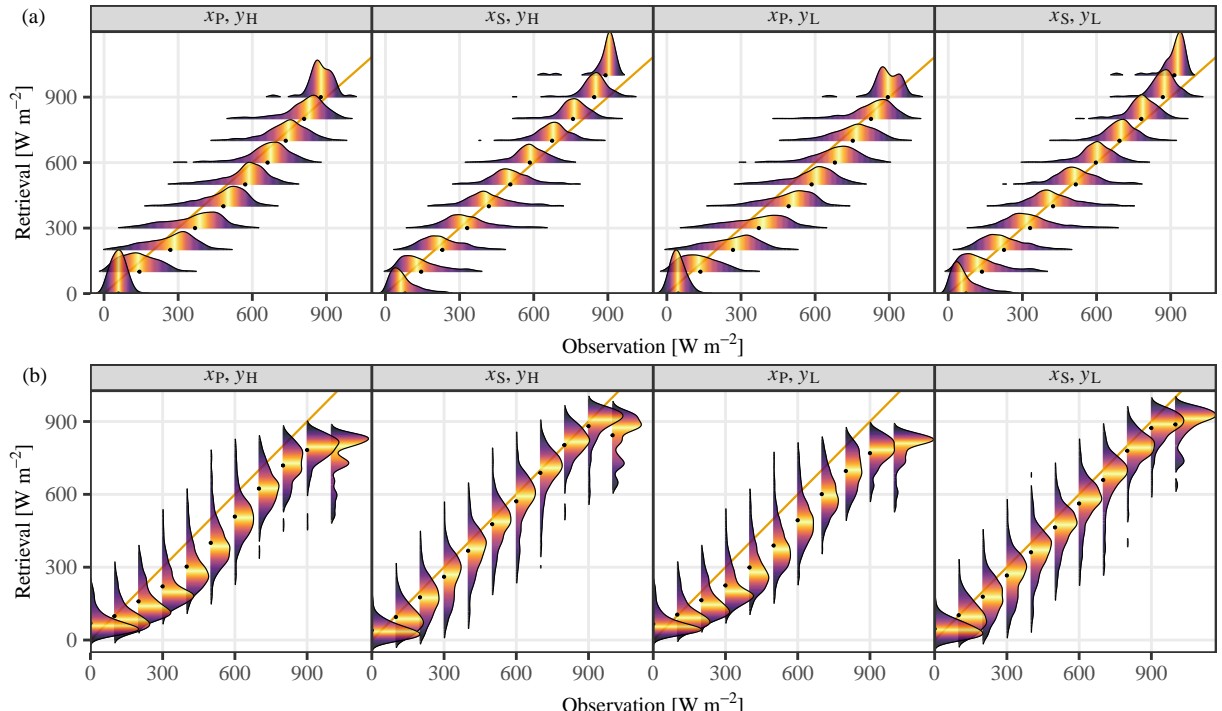

**Figure 6.** The conditional PDFs of (a) observation given retrieval and (b) retrieval given observation, for different $x_P$, $x_S$, $y_H$, and $y_L$ combinations. The colors indicate quantiles of the respective conditional PDF, with brighter ones being closer to the median.

$\mathbb{E}(Y|X)]^2$, $\mathbb{E}_X[\mathbb{E}(Y|X) - \mathbb{E}(Y)]^2$, $\mathbb{E}_Y[Y - \mathbb{E}(X|Y)]^2$, and $\mathbb{E}_Y[\mathbb{E}(X|Y) - \mathbb{E}(X)]^2$, respectively. These aspects of product quality are summarized and defined in Table 2, alongside their relevance to the distribution-oriented validation.

Table 3 lists the results of bias–variance decomposition of MSE, based on different combinations of $x_P$, $x_S$, $y_H$, and $y_H$. In
terms of the variances, the standard deviation of $y_H$ is 243 W m$^{-2}$, and compared to that, $y_L$ is overdispersed with a standard deviation of 254 W m$^{-2}$. As for the retrievals, $x_S$ is even more overdispersed, whereas $x_P$ is underdispersed. On this point, the variance of $x_S$ is closer to that of $y_L$ and the variance of $x_P$ is closer to that of $y_S$, which implies that $y_L$ exaggerates the performance of $x_S$ but understates that of $x_P$. In terms of association, i.e., the $\text{Cov}(X,Y)$ column, the highest value ($248^2$ W$^2$ m$^{-4}$) is seen when the statistical retrievals are validated using low-accuracy observations, whereas the lowest value ($226^2$ W$^2$ m$^{-4}$)
is seen when the physical retrievals are validated using high-accuracy observations. For both physical and statistical retrievals, the results are consistent—the low-accuracy observations tend to yield overconfident results by exaggerating the association. In terms of the unconditional bias, i.e., the $[\mathbb{E}(X) - \mathbb{E}(Y)]^2$ column, the opposite is true, in that, the low-accuracy observations return larger biases, announcing the products worse than they actually are, by about 3 W m$^{-2}$, which is quite significant when considering it in the light of currently available high accuracy satellite-retrieved irradiance datasets Yang and Bright (2020).
Overall, Table 3 strongly evidences that low-accuracy observations should not be used to validate gridded irradiance.





**Table 2.** The definitions and relevance of different aspects of product quality to distribution-oriented validation (Murphy, 1993).

| Aspect | Definition | Rel. distribution | Metric |
|---|---|---|---|
| Association | Overall strength of linear relationship between individual pairs of retrieval and observation | $f(x,y)$ | $\text{Cov}(X,Y)$ |
| Uncond. bias | Directional difference between mean retrieval and mean observation | $f(x)$ and $f(y)$ | $[\mathbb{E}(X) - \mathbb{E}(Y)]^2$ |
| Calibration | (Also known as reliability or type-1 conditional bias) Correspondence between conditional mean observation and conditioning retrieval, averaged over all retrievals | $f(y|x)$ and $f(x)$ | $\mathbb{E}_X[X - \mathbb{E}(Y|X)]^2$ |
| Resolution | Difference between conditional mean observation and unconditional mean observation, averaged over all retrievals | $f(y|x)$ and $f(x)$ | $\mathbb{E}_X[\mathbb{E}(Y|X) - \mathbb{E}(Y)]^2$ |
| Type-2 cond. bias | (Also known as discrimination 1) Correspondence between conditional mean retrieval and conditioning observation, averaged over all observations | $f(x|y)$ and $f(y)$ | $\mathbb{E}_Y[Y - \mathbb{E}(X|Y)]^2$ |
| Discrimination | (Also known as discrimination 2) Difference between conditional mean retrieval and unconditional mean retrieval, averaged over all observations | $f(x|y)$ and $f(y)$ | $\mathbb{E}_Y[\mathbb{E}(X|Y) - \mathbb{E}(X)]^2$ |

**Table 3.** Results of bias–variance decomposition of MSE, cf. Eq. (5), based on different combinations of $x_\text{P}$, $x_\text{S}$, $y_\text{H}$, and $y_\text{H}$. The metrics are written as exponentiations, such that all bases have the unit of $\text{W m}^{-2}$.

| Obs. | Ret. | $\text{MSE}(X,Y)$ | $\mathbb{V}(X)$ | $\mathbb{V}(Y)$ | $\text{Cov}(X,Y)$ | $[\mathbb{E}(X) - \mathbb{E}(Y)]^2$ |
|---|---|---|---|---|---|---|
| $y_\text{H}$ | $x_\text{P}$ | $107^2$ | $226^2$ | $243^2$ | $226^2$ | $58^2$ |
| $y_\text{H}$ | $x_\text{S}$ | $92^2$ | $258^2$ | $243^2$ | $243^2$ | $18^2$ |
| $y_\text{L}$ | $x_\text{P}$ | $114^2$ | $226^2$ | $254^2$ | $231^2$ | $61^2$ |
| $y_\text{L}$ | $x_\text{S}$ | $93^2$ | $258^2$ | $254^2$ | $248^2$ | $21^2$ |

Moving on to Table 4, which shows the results of calibration–refinement decomposition of MSE, more interesting insights can be gained. First, the physically retrieved product is less calibrated than the statistical one, which agrees with the earlier visual assessment. Recall that for two sets of retrievals, the one with a smaller value of calibration should be preferred. On this point, the calibration terms corresponding to $x_\text{P}$—see the $\mathbb{E}_X[X - \mathbb{E}(Y|X)]^2$ column—are higher than those of $x_\text{S}$, rendering $x_\text{P}$ inferior. In terms of resolution, i.e., the $\mathbb{E}_X[\mathbb{E}(Y|X) - \mathbb{E}(Y)]^2$ column, the choice of observations appears to have a larger impact on the validation results than the products themselves, which is a bit surprising. The resolutions of both $x_\text{P}$ and $x_\text{S}$ are $\sim 227^2 \text{ W}^2 \text{ m}^{-4}$ when evaluated against $y_\text{H}$, and are about $\sim 237^2 \text{ W}^2 \text{ m}^{-4}$ when evaluated against $y_\text{L}$. Since for a given set of observations, $\mathbb{E}(Y)$ takes a fixed value, the similarity must be attributed to $\mathbb{E}(Y|X)$, and one can conclude that the resolving power of $x_\text{P}$ and $x_\text{S}$ are comparable.





**Table 4.** Same as Table 3, but for calibration–refinement decomposition of MSE, cf. Eq. (6).

| Obs. | Ret. | $\mathrm{MSE}(X,Y)$ | $\mathbb{V}(Y)$ | $\mathbb{E}_X[X - \mathbb{E}(Y|X)]^2$ | $\mathbb{E}_X[\mathbb{E}(Y|X) - \mathbb{E}(Y)]^2$ |
|---|---|---|---|---|---|
| $y_\mathrm{H}$ | $x_\mathrm{P}$ | $107^2$ | $243^2$ | $62^2$ | $227^2$ |
| $y_\mathrm{H}$ | $x_\mathrm{S}$ | $92^2$ | $243^2$ | $36^2$ | $228^2$ |
| $y_\mathrm{L}$ | $x_\mathrm{P}$ | $114^2$ | $254^2$ | $66^2$ | $237^2$ |
| $y_\mathrm{L}$ | $x_\mathrm{S}$ | $93^2$ | $254^2$ | $30^2$ | $238^2$ |

For the likelihood–base rate decomposition of MSE, its results are tabulated in Table 5. In terms of the type-2 conditional bias, the statistically retrieved product again shows superiority, regardless of whether the high- or low-accuracy observations are used for validation. It is worth noting that the type-2 conditional bias of $x_\mathrm{S}$ appears higher when validated using $y_\mathrm{L}$ than using $y_\mathrm{H}$, which contrasts the former case, where the calibration (i.e., type-1 conditional bias) of $x_\mathrm{S}$ appears higher when validated using $y_\mathrm{H}$ than using $y_\mathrm{L}$. As for discrimination, the difference in validation results across products is now larger than the difference across observation choices, which also contradicts the finding from Table 4.

**Table 5.** Same as Table 3, but for likelihood–base rate decomposition of MSE, cf. Eq. (7).

| Obs. | Ret. | $\mathrm{MSE}(X,Y)$ | $\mathbb{V}(X)$ | $\mathbb{E}_Y[Y - \mathbb{E}(X|Y)]^2$ | $\mathbb{E}_Y[\mathbb{E}(X|Y) - \mathbb{E}(X)]^2$ |
|---|---|---|---|---|---|
| $y_\mathrm{H}$ | $x_\mathrm{P}$ | $107^2$ | $226^2$ | $72^2$ | $212^2$ |
| $y_\mathrm{H}$ | $x_\mathrm{S}$ | $92^2$ | $258^2$ | $25^2$ | $243^2$ |
| $y_\mathrm{L}$ | $x_\mathrm{P}$ | $114^2$ | $226^2$ | $80^2$ | $211^2$ |
| $y_\mathrm{L}$ | $x_\mathrm{S}$ | $93^2$ | $258^2$ | $30^2$ | $243^2$ |

To give a summary of the quantitative assessment, a linear ranking method (see Alvo and Yu, 2014) is applied to the above results. There are four combinations of $x_\mathrm{P}$, $x_\mathrm{S}$, $y_\mathrm{H}$, and $y_\mathrm{H}$; the mean rank of the $i$th combination, as denoted by $m_i$, is:

$$m_i = \sum_{j=1}^{4!} \frac{n_j v_j(i)}{n}, \tag{8}$$

where $v_j$ with $j = 1, 2, \ldots, 4!$ represent all possible rankings of the four combinations, $n_j$ is the frequency of occurrence of these rankings, $n = \sum n_j$ is the total number of ranking exercises, and $v_j(i)$ denotes the score of the $i$th combination in ranking $j$. Table 6 shows the ranking of different combinations, together with their mean ranks computed using Eq. (8). It can be seen that there are four rankings, namely, $v_1(i) = (3, 1, 4, 2)$, $v_2(i) = (4, 2, 3, 1)$, $v_3(i) = (3, 2, 4, 1)$, and $v_4(i) = (4, 3, 2, 1)$; the corresponding $n_j$'s are 3, 1, 2, and 1, respectively. The foremost finding is that the rankings based on various quantification metrics are not unified, which again highlights the pitfall of measure-oriented validation. Next, the statistically retrieved irradiance, in this case, outperforms the physically retrieved ones, which is to be further investigated below. Last but not least, when low-accuracy observations are used to validated satellite-retrieved products, it can offset the results in both directions—i.e., $y_\mathrm{L}$ exaggerates the performance of $x_\mathrm{S}$ but understates that of $x_\mathrm{P}$—which is not desirable in any case. Before





we end this section, it is worth mentioning that variance plays an important role in validation. The variance ratio between observation and retrieval is further linked to calibration, type-2 conditional bias, and discrimination. Interested readers are referred to Mayer and Yang (2023) for further reading.

**Table 6.** Ranking of validation results in terms of different aspects of product quality.

| Aspects | $y_H, x_P$ | $y_H, x_S$ | $y_L, x_P$ | $y_L, x_S$ |
|---|---|---|---|---|
| Accuracy (i.e., MSE) | 3 | 1 | 4 | 2 |
| Association | 4 | 2 | 3 | 1 |
| Uncond. bias | 3 | 1 | 4 | 2 |
| Calibration | 3 | 2 | 4 | 1 |
| Resolution | 4 | 3 | 2 | 1 |
| Type-2 cond. bias | 3 | 1 | 4 | 2 |
| Discrimination | 3 | 2 | 4 | 1 |
| Mean rank | 3.29 | 1.71 | 3.57 | 1.43 |

### 3.3 Validation of hourly and daily averages

Among the various factors that influence the quality of satellite-retrieved irradiance, the three-dimensional (3D) effects of clouds, including the parallax effect, arguably constitute the most significant source of error. This phenomenon manifests when cloud presence in either the sun-to-surface path or the surface-to-satellite path fails to coincide, leading to erroneous sky condition identification (Huang et al., 2019). Such misidentification poses substantial contamination risks to retrieval accuracy and thus validation outcomes, prompting some researchers to advocate for the exclusion of retrieval–observation pairs exhibiting differences exceeding three standard deviations from the mean (Wang and Pinker, 2009). An alternative approach is based on the well-documented fact that the 3D effect of clouds rapidly decreases as the temporal resolution of retrieval gets coarse (Huang et al., 2016). This has led to widespread adoption of hourly and daily product validation in the literature (e.g., Shi et al., 2023; Huang et al., 2023). In accordance with this methodological convention, we have aggregated the original 15-min data into both hourly and daily resolutions in this section, with sample numbers of 3740 and 355, respectively.

In accordance with the presentation style of Table 1, the overall biases and accuracies of hourly and daily samples are listed in Table 7. Comparing these tables reveals that the accuracy improves as the temporal resolution of the samples decreases. In contrast, the bias becomes larger after averaging. Particularly for daily samples, the bias of $x_P$ shoots over $-19\%$, unequivocally demonstrating fundamental algorithmic deficiencies, which motivate site-specific bias correction. That said, given that aggregation should normally not affect bias, the differences between the 15-min and daily MBE come from the fact that days are shorter in winter—i.e., the proportion of the 15-min winter samples is lower than the proportion of the daily winter samples. The fact that the MBE decreased to a higher underestimation for the daily mean GHI in itself shows that the underestimation is higher in winter than in summer, which is shown in more detail in a later section.



Figure 7 shows the joint PDFs of hourly and daily samples, under different combinations of $\overline{x}_{\mathrm{P}}$, $\overline{x}_{\mathrm{S}}$, $\overline{y}_{\mathrm{H}}$, and $\overline{y}_{\mathrm{L}}$, with bars annotating averages. The joint PDFs of the hourly samples exhibit notable consistency with those of the 15-min samples from Fig. 4, e.g., the consistent underestimation and nonlinearity in $\overline{x}_{\mathrm{P}}$ seen earlier are again visible. The joint PDFs of the daily samples reveal a pronounced density peak near 450 $\mathrm{W\,m^{-2}}$, indicative of prevailing clear-sky conditions at the validation location. Of particular concern is that the scatter of the daily samples of $\overline{x}_{\mathrm{S}}$ only packs very loosely around the identity line, especially over the low irradiance range of 100–300 $\mathrm{W\,m^{-2}}$, which contradicts earlier reports of similar Fengyun irradiance products, cf. Fig. 5 in Huang et al. (2023); Fig. 5 in Shi et al. (2023). This discrepancy potentially suggests the influence of winter snow cover on retrieval accuracy within the Dwa Köppen climate regime; this hypothesis is to be further examined in Section 3.5.

**Table 7.** Same as Table 1, but for hourly and daily samples.

| | Hourly samples | | | | Daily samples | | | |
|---|---|---|---|---|---|---|---|---|
| | MBE (nMBE) | | RMSE (nRMSE) | | MBE (nMBE) | | RMSE (nRMSE) | |
| | $\overline{x}_{\mathrm{P}}$ | $\overline{x}_{\mathrm{S}}$ | $\overline{x}_{\mathrm{P}}$ | $\overline{x}_{\mathrm{S}}$ | $\overline{x}_{\mathrm{P}}$ | $\overline{x}_{\mathrm{S}}$ | $\overline{x}_{\mathrm{P}}$ | $\overline{x}_{\mathrm{S}}$ |
| $\overline{y}_{\mathrm{H}}$ | $-59\,(-16.0\%)$ | $-20\,(-5.3\%)$ | $98\,(26.6\%)$ | $76\,(20.7\%)$ | $-67\,(-19.4\%)$ | $-24\,(-6.9\%)$ | $86\,(24.9\%)$ | $58\,(16.8\%)$ |
| $\overline{y}_{\mathrm{L}}$ | $-61\,(-16.5\%)$ | $-21\,(-5.8\%)$ | $104\,(28.2\%)$ | $77\,(20.9\%)$ | $-68\,(-19.7\%)$ | $-25\,(-7.3\%)$ | $89\,(25.5\%)$ | $58\,(16.6\%)$ |

### 3.4 Validation results after bias correction

Within the domain of solar energy meteorology, the imperative for bias correction in satellite-retrieved irradiance arises from a fundamental economic consideration: The precision of mean irradiance estimates bears direct consequence upon the financial viability and projected returns of solar energy projects (Yang et al., 2022b). This location-specific bias correction procedure is better known as site adaptation, which has attracted much research effort in the past decade (e.g., Zainali et al., 2024; Yang and Gueymard, 2021; Polo et al., 2016). Among various statistical and machine learning site-adaptation methods, quantile mapping stands as a classic option. More importantly, quantile mapping is more amenable than linear regression when nonlinearity is present in the data, such as the case of $x_{\mathrm{P}}$. Mathematically, quantile mapping is simply

$$\widehat{x} = \widehat{F}_Y^{-1}\left[\widehat{F}_X(x)\right], \tag{9}$$

where $\widehat{F}_Y(\cdot)$ and $\widehat{F}_X(\cdot)$ are empirical cumulative distribution functions (ECDFs) of observation and retrieval, respectively; $x$ is an arbitrary retrieval value; and $\widehat{x}$ is the corrected value.

Certainly, $\widehat{F}_Y(\cdot)$ and $\widehat{F}_X(\cdot)$ need to be trained before applying quantile mapping to new data points. To that end, the 15-min samples are randomly split into two groups, namely, A and B. After the samples in B are quantile mapped using the $\widehat{F}_Y(\cdot)$ and $\widehat{F}_X(\cdot)$ trained using A, the two groups switch, such that the bias-corrected versions of $x_{\mathrm{P}}$ and $x_{\mathrm{S}}$ has the same number of samples as the original data. The updated MBEs and RMSEs are provided in Table 8. It can be seen that the bias has





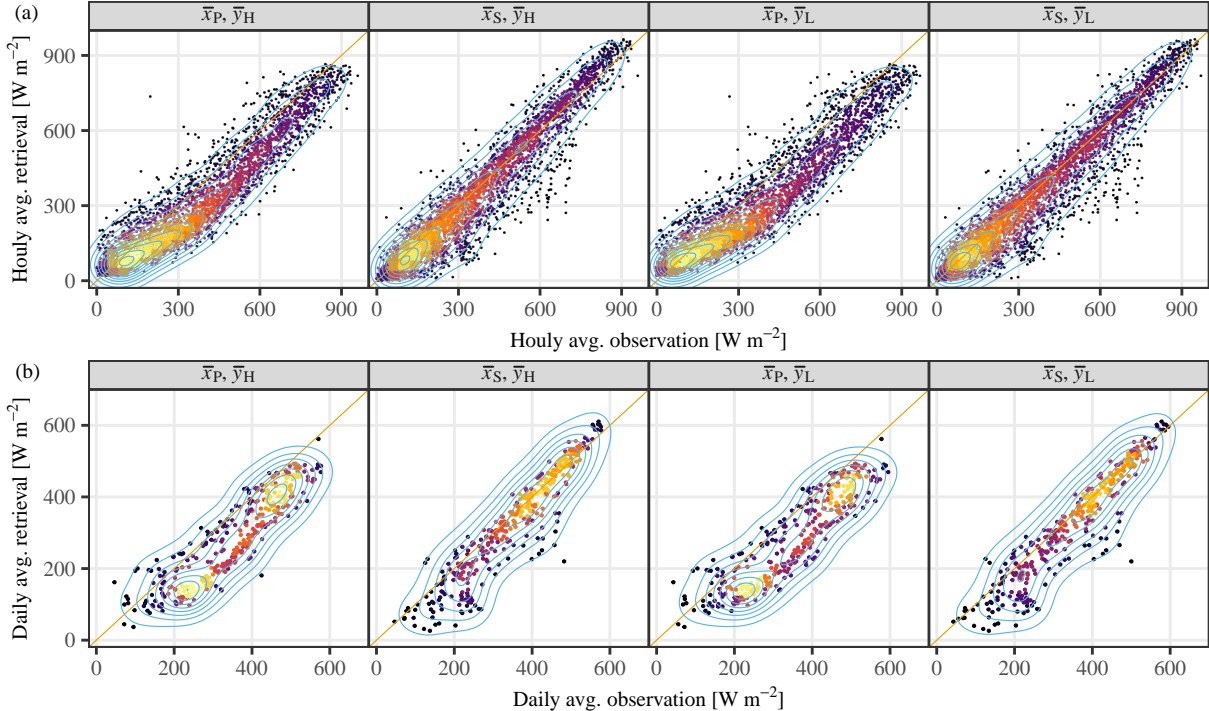

**Figure 7.** The joint PDFs of (a) hourly averages and (b) daily averages of retrievals and observations.

been effectively eliminated, and thus translated to improved accuracy, in all cases. We further test the effect of combining the
465 physical and statistical products by taking the arithmetic mean of the two time series; a substantial reduction in RMSE implies
the complementarity of the two products. Nonetheless, despite that the bias-corrected samples have no bias with respect to both
$y_H$ and $y_L$, the RMSEs against $y_L$ are higher across board, indicating that using low-accuracy observations as reference during
validation may return underconfident results, by about 0.8% in this particular case.

**Table 8.** Same as Table 1, but for the bias-corrected $x_P$ and $x_S$ after quantile mapping, which are denoted as $\widehat{x}_P$ and $\widehat{x}_S$. Additionally, the
results of a simple combination of $\widehat{x}_P$ and $\widehat{x}_S$ are shown in the $(\widehat{x}_P + \widehat{x}_S)/2$ columns, demonstrating the complementarity of the two products.

| | MBE (nMBE) | | | RMSE (nRMSE) | | |
|---|---|---|---|---|---|---|
| | $\widehat{x}_P$ | $\widehat{x}_S$ | $(\widehat{x}_P + \widehat{x}_S)/2$ | $\widehat{x}_P$ | $\widehat{x}_S$ | $(\widehat{x}_P + \widehat{x}_S)/2$ |
| $y_H$ | 0 (0.0%) | 1 (0.1%) | 0 (0.1%) | 90 (24.2%) | 87 (23.5%) | 74 (20.1%) |
| $y_L$ | 0 (0.0%) | 1 (0.1%) | 0 (0.1%) | 95 (25.4%) | 90 (24.1%) | 78 (20.8%) |





### 3.5 Noticeable algorithmic limitations during winter

Hitherto, our validation has proceeded through comprehensive analyses of all available samples. While these aggregate results adequately characterize the general performance attributes of both products, they necessarily obscure certain extreme cases where algorithmic performance degrades substantially. A more penetrating examination of these specific conditions promises to reveal important limitations in the underlying retrieval algorithms that might otherwise escape notice. For instance, all of those severely underestimated daily samples of $\overline{x}_\mathrm{S}$ in Fig. 7 (b), with $\overline{x}_\mathrm{S} - \overline{y}_\mathrm{H} < -80\ \mathrm{W\,m^{-2}}$, come from winter months,

which indicates a potential algorithmic limitation of Heliosat-2 during winter, with prolonged snow cover being typical over November to March for the Dwa climate.

The most transparent method for diagnosing temporal data lies in direct visual examination through time series plotting. In this regard, Fig. 8 shows the satellite-observed reflectance ($\rho$), the corresponding dynamic range ($\rho_\mathrm{H}$ and $\rho_\mathrm{L}$), cloud index ($\nu$), and clear-sky index ($\kappa$), over the winter months; the reader is referred to Appendix A for the relationship among these

480 variables under the Heliosat-2 framework. Several insights are revealed. First, the satellite-observed reflectance exhibits an abrupt jump in late November and stays high until mid-March, which may be attributed to the long-lasting snow cover at the QIQ station. To verify this hypothesis, data from the United States National Ice Center's (NIC's) Interactive Multisensor Snow and Ice Mapping System (IMS) is considered; Fig. 9 (a) exemplifies the original data, which is in NetCDF format and polar stereographic projection. Figure 9 (b) displays the time series of the IMS surface value at the QIQ station, over November and

485 December 2024, which matches well with the elevated $\rho$ time series, thus confirming the hypothesis.

The next insight emerges from the monthly constant dynamic range employed by this version of the Heliosat-2 algorithm. In December, for instance, the lower bound $\rho_\mathrm{L}$ is underdetermined, resulting in overestimated $\nu$ and underestimated $\kappa$, as depicted in Figs. 8 (b) and (c). In contrast, $\rho_\mathrm{L}$ in February is overdetermined, resulting in numerous $\nu < -0.2$ cases, which are unphysical. According to the Heliosat-2 computation, these cases would return $\kappa = 1.2$, which is, again, extremely rare in

reality. Overall, as evidenced by Fig. 9 (c), which plots the $(x_\mathrm{S}, y_\mathrm{H})$ scatter for all days with snow cover, the snow cover causes this version of the Heliosat-2 algorithm to severely underestimate the irradiance. Stated in another way, one may interpret this phenomenon as a case of cloud misidentification, in that, the algorithm is unable to distinguish the bright surface due to snow cover from clouds. To remedy this algorithmic pitfall, the bounds of dynamic range ought to be determined in a rolling fashion, which has been emphasized by Perez et al. (2002) in a very early publication. The difficulty, however, is one of computation—

the iterative method for dynamic range determination for the full-disk area is too taxing, which promotes investigations into advanced sorting algorithms.

Whereas the algorithm limitations of the statistical product can be identified, the physical product is much more difficult to diagnose, given the complexity of the physical retrieval process. Therefore, only a simple visualization that reveals some product quality issues in winter is made, alongside a discussion on potential causes. Figure 10 shows the time series plot of

500 $x_\mathrm{P}$, $y_\mathrm{H}$, and the McClear clear-sky irradiance in February 2025. Although the overall underestimation of product is evident, it is fairly consistent throughout the month. From the plot of February 8, 9, and 13, one may conclude that the product is able to capture the effect of clouds on irradiance to a good extent, resulting in ramp rates similar to the observations. Instead, it is





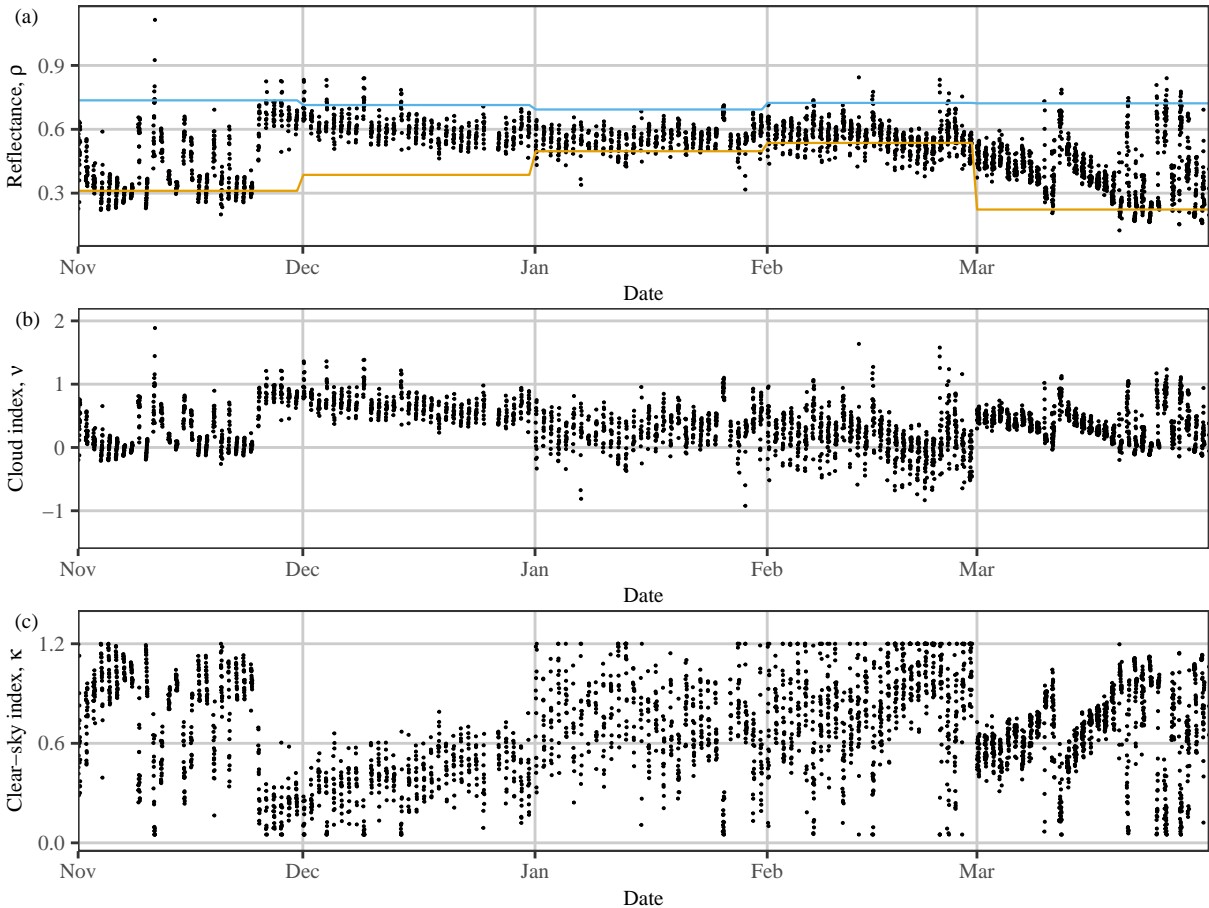

**Figure 8.** (a) The satellite-observed reflectance (black dots) and dynamic range (colored lines) from November 2024 to March 2025, with the corresponding (b) cloud index, $\nu$, and (c) clear-sky index, $\kappa$.

the understimation of the clear-sky irradiance, see e.g., February 4, 12, and 22, that affects the product quality. As clear-sky irradiance is primarily affected by the extraterrestrial irradiance, the sun–earth geometry, aerosols, and water vapors, with the former two being trivial to obtain, the underestimation is likely due to the imprecise information of the latter two, which echoes our earlier hypothesis in Section 3.2. This version of the NSMC's algorithm adopts an "indirect" approach that requires the AGRI observed channel reflectances as main inputs with little reliance on level-2 products, such as aerosol or water vapor. To that end, the inaccurate aerosol climatology used during retrieval might be the root cause of the systematic underestimation.





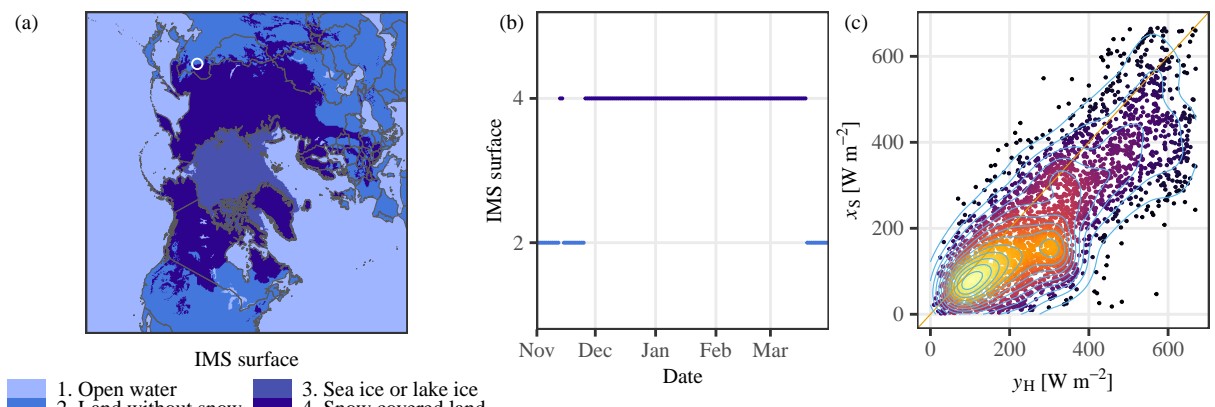

**Figure 9.** (a) The NIC IMS map (in polar stereographic projection) of November 24, 2024, with the QIQ station marked with a white circle. (b) The time series of the IMS surface value at the QIQ station, from November 2024 to March 2025. (c) The scatter plot of $x_S$ against $y_H$ for all days with snow cover.

## 4 Conclusion

Validation of satellite-retrieved irradiance is ubiquitously performed in comparative studies concerning the quality of several products (e.g., Salazar et al., 2020; Marchand et al., 2018), as well as those studies investigating the change in radiation budget of the earth (e.g., Sanchez-Lorenzo et al., 2017; Zhang et al., 2015). With the validation results, various scientific assertions and conclusions are made. A critical yet frequently overlooked consideration concerns the reliability of observed inter-product variations or temporal radiation trends, which may be systematically distorted by the inherent limitations of the ground-based reference data employed in the validation process. To that end, this work formally investigates this issue by leveraging both high- and low-accuracy radiometric observations during the validation of two satellite-retrieved irradiance products for a location in a monsoon-influenced continental climate.

The contribution of this work is three-fold. First, irradiance observations from a rarely reported climate regime (i.e., the Dwa Köppen climate) are presented, alongside two gridded irradiance products from the latest Chinese FY-4B weather satellite. Through rigorous analysis, we have identified a fundamental limitation in the operational physical retrieval algorithm implemented by NSMC: Its systematic underestimation of irradiance under clear-sky conditions reveals significant shortcomings in algorithmic maturity and implementation, particularly in terms of the quality of those atmospheric parameters intermediate to irradiance retrieval, such as aerosol optical depth or water vapor. Conversely, the statistical retrieval algorithm yields demonstrably superior results compared to the official NSMC product, at last for the geographical location under examination. However, the Heliosat-2 algorithm suffers substantial performance degradation under snow-covered conditions, in which the high surface albedo observed by the imager might be wrongfully attributed to clouds.

The second contribution resides in the novel application of the distribution-oriented validation framework, which was originally proposed to verify weather forecasts. Contrasting the measure-oriented validation, which is limited by the choice of those





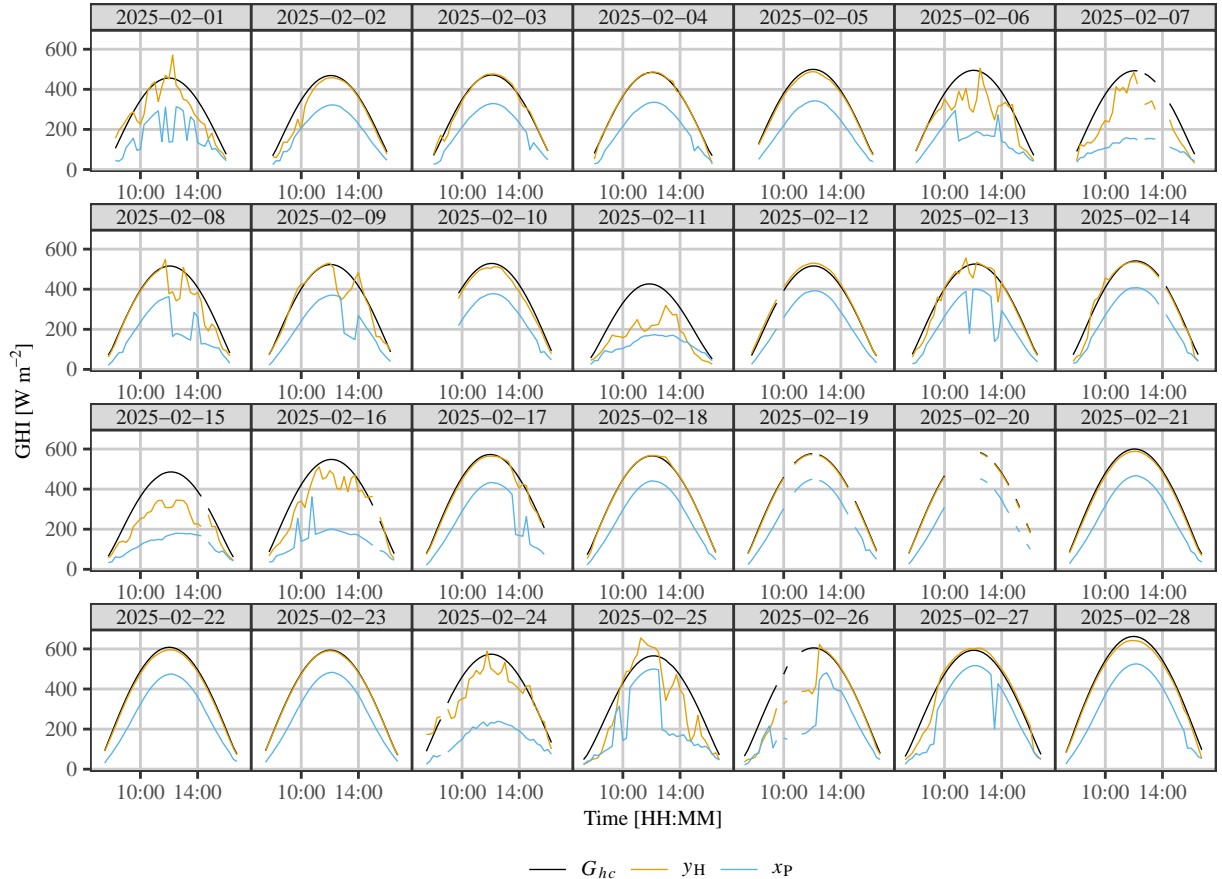

**Figure 10.** The time series plot of $x_\mathrm{P}$ and $y_\mathrm{H}$ in February 2025, alongside McClear GHI ($G_{hc}$).

often confounding performance measures, the visual and quantitative assessments of the joint, marginal, and conditional distri-
butions of retrieval and observation allow a more comprehensive understanding of various aspects of product quality, including
bias, accuracy, association, calibration, resolution, and discrimination (cf. Tables 3–5). The main takeaway is that even if a
certain product is validated to be strictly inferior in terms of bias and accuracy, it may still possess advantages in other aspects
of quality (cf. Table 6). This complementarity between two or more products promotes combining or merging, as demonstrated
in Table 8, which has led to a 4% improvement in accuracy in this particular case.

Third, in regard to the quality of ground truth, the accumulated empirical evidence demonstrates conclusively that employ-
ing low-accuracy observations for irradiance retrieval validation introduces measurable deviations from what may be properly
regarded as the "true"' values—that is to say, those values obtained through high-accuracy observational methods. Given the
large sample size, these discrepancies represent actual differences in the validation outcomes rather than artifacts of mea-
surement uncertainty. It must be highlighted that the deviations from the "true" value are not unidirectional, i.e., using the



low-accuracy observations can yield both overconfident and underconfident results, depending on the particular metric un-
der consideration. Moreover, the magnitude of the deviations caused by the low-accuracy observations is quite substantial,
reaching several $\mathrm{W\,m^{-2}}$ or a few percent. Indeed, numerous prior conclusions, such as the assertion that "satellite products
overestimated radiation by approximately 10 $\mathrm{W\,m^{-2}}$" by Zhang et al. (2015), rest upon differentials of comparable scale.
This epistemic circumstance necessitates rigorous methodological caution: Any validation exercise employing low-accuracy

observations must interpret apparent results with appropriate epistemological restraint, recognizing the substantial uncertainty
introduced by the observational limitations themselves.

*Code and data availability.* A total of four datasets are involved in this work, among which two are ground-based radiometric observations
and the other two are satellite-retrieved irradiance from L1 Fengyun-4B products. The high-accuracy ground-based radiometric observations
can be obtained from the official website of the Baseline Surface Radiation Network (https://bsrn.awi.de/data/data-retrieval-via-ftp/). The

low-accuracy ground-based radiometric observations are not publicly available but were obtained by contacting the China Meteorological
Administration. L1 Fengyun-4B products are available from the FENGYUN Satellite Data Service (https://satellite.nsmc.org.cn/DataPortal/
en/data/structure.html).All code will be made available upon request.

**Appendix A: The Heliosat-2 algorithm**

In a nutshell, the Heliosat-2 algorithm (Rigollier et al., 2004) first estimates the cloud index, $\nu$, from the satellite-observed

reflectance, and then converts it to the clear-sky index, $\kappa$, via

$$
\kappa = \begin{cases}
1.2, & \text{if } \nu < -0.2; \\
a_0 + a_1\nu, & \text{if } -0.2 \leq \nu < \alpha; \\
b_0 + b_1\nu + b_2\nu^2, & \text{if } \alpha \leq \nu < 1.1; \\
0.05 & \text{if } 1.1 \leq \nu;
\end{cases}
\tag{A1}
$$

where model parameters $\alpha = 0.9$, $a_0 = 1.008$, $a_1 = -0.96$, $b_0 = 2.8935$, $b_1 = -5.1699$, and $b_2 = 2.3499$ are determined em-
pirically from data, in this case, observations from 38 ground-based stations in China (Huang et al., 2023). On the other hand,
the determination of $\nu$ follows

$$
\nu = \frac{\rho - \rho_\mathrm{L}}{\rho_\mathrm{H} - \rho_\mathrm{L}},
\tag{A2}
$$

where $\rho$ is the satellite-observed reflectance after sun-to-surface and surface-to-satellite geometric corrections; $\rho_\mathrm{L}$ and $\rho_\mathrm{H}$ are
the lower and upper bounds of the dynamic range derived from an adequate collection of $\rho$ values, e.g., over a time period
or over an area, see below. Physically, $\rho_\mathrm{L}$ corresponds to the expected clear-sky reflectance, whereas $\rho_\mathrm{H}$ corresponds to the
expected reflectance of the brightest clouds.

In this version of the Heliosat-2 algorithm, $\rho_\mathrm{H}$ is first determined. More specifically, for each month, all samples of $\rho$ with a
zenith angle $Z < 80°$, over China, are gathered and sorted; the 95% quantile is then set as $\rho_\mathrm{H}$. In other words, the $\rho_\mathrm{H}$ value for



all of China is the same for a given month. In contrast, the determination of $\rho_{\mathrm{L}}$ is pixel-specific, and an iterative algorithm is used. Denoting the sorted samples of $\rho$ from a specific pixel and a specific month with $Z < 75°$ as $\rho_i$, with $i = 1, 2, \ldots, n^{(0)}$, a threshold $t^{(1)}$ is obtained via

$$t^{(1)} = \sum_{i=1}^{n^{(0)}} \rho_i + 0.035\rho_{\mathrm{H}}. \tag{A3}$$

Subsequently, all $\rho_i$'s that exceed threshold $t^{(1)}$ are removed, which results in $n^{(1)}$ remaining samples. In the second iteration, one proceeds with calculating

$$t^{(2)} = \sum_{i=1}^{n^{(1)}} \rho_i + 0.035\rho_{\mathrm{H}}, \tag{A4}$$

and $\rho_i$'s that exceed threshold $t^{(2)}$ are removed. This iteration terminates when all remaining samples of $\rho_i$ are below the threshold, and their average is used as $\rho_{\mathrm{L}}$.

*Author contributions.* **Yu Chen**: Conceptualization, Methodology, Software, Validation, Investigation, Writing – original draft. **Dazhi Yang**: Conceptualization, Methodology, Software, Resources, Writing – original draft, Visualization, Supervision, Project administration, Funding acquisition. **Chunlin Huang**: Data curation, Formal analysis, Investigation, Writing – review & editing. **Hongrong Shi**: Data curation, Formal analysis, Writing – review & editing. **Adam R. Jensen**: Formal analysis, Visualization, Writing – review & editing. **Xiang'ao Xia**: Validation, Project administration, Writing – review & editing. **Yves-Marie Saint-Drenan**: Methodology, Visualization, Validation, Writing – review & editing. **Christian A. Gueymard**: Validation, Writing – review & editing. **Martin János Mayer**: Formal analysis, Investigation, Writing – review & editing. **Yanbo Shen**: Methodology, Supervision, Writing – review & editing.

*Competing interests.* The authors declare that they have no known competing financial interests or personal relationships that could have appeared to influence the work reported in this paper.

*Acknowledgements.* This work is supported in part by the National Natural Science Foundation of China (No. 42375192).



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
