# Peer review of "Validating physical and semi-empirical satellite-based irradiance retrievals using high- and low-accuracy radiometric observations in a monsoon-influenced continental climate"

_EGUsphere, 2025_

## Author Response (AR1)

**Response to reviewers**

We would like to thank all reviewers for their valuable time and feedback. The reviewers' comments are in *Times (italic and bold)*, our response is in Red Arial.

*Reviewer #1:*

*This paper presents a critically important investigation into whether low-accuracy radiometric observations can substitute high-accuracy ones for validating satellite-derived irradiance. The authors develop a rigorous, distribution-oriented validation framework to systematically compare physical and statistical satellite retrievals against both types of ground observations. Using a novel dataset from a new BSRN station and Fengyun-4B satellite retrievals, the study delivers a crucial finding: employing low-accuracy references introduces significant risks, as resulting discrepancies are comparable to commonly accepted error margins in the field. This work provides both a methodological advancement and an essential caution for satellite product validation practices. Here, I have some additional points that I think should be considered before publication. Line numbers refer to the tracked revised manuscript.*

Thank you for the accurate summary of our work and the confirmation on the contribution. We have revised the work thoroughly according to your comments.

*1. The abstract should present the key quantitative findings more explicitly, specifically the magnitude of discrepancy in $W\ m^{-2}$ between different validation references.*

Thank you for your comment. We could include some numerical results in the abstract to make it more explicit. For instance, the sentence could read "accepted margins of error or improvement (up to 7 W m$^{-2}$ or 1.5%)." However, we feel that stating the exact number greatly limits the finding. That is why we stated the finding in a more general way, that is, "commonly accepted margins of error or improvement (approximately several W m$^{-2}$ or a few percent) upon which numerous scientific assertions depend." Also, if we change to the exact numbers, the sentence no longer reads—how can numerous scientific assertions depend on 7 W m$^{-2}$ or 1.5%?

*2. The introduction cites multiple works by the corresponding author. To better situate the study within the broader field, it would be beneficial to also incorporate key perspectives and findings from other research groups.*

Thank you for your comment. We have included several works from other research groups in the introduction.

*3. Line 134: Please note that the China Meteorological Administration (CMA) currently maintains over 90,000 weather stations. The text should be updated to reflect this more accurate and recent figure.*

Thank you for this important correction. We have changed the number during the revision.

*4. Line 177: The phrasing "shows stronger stability" could be improved for technical precision. It is recommended to rephrase it to state that "FY-4B demonstrates superior calibration stability compared to FY-4A."*

Thank you for your suggestion. We have included the above sentence into the corresponding position of text.

*5. Figure 2 caption: The current description is insufficient. Please provide a self-contained explanation for each sub-figure so that readers can understand them without needing to consult the referenced literature.*

Thank you for your suggestion. We have enhanced the caption of Fig. 2 in the revised version.

*6. Color bars are missing in Figure 3 and other probability density function (PDF) plots. These must be added for correct interpretation of the data.*

Thank you for your comment. The colors, as indicated in the figure caption, simply refer to the density of points in the neighborhood. Their exact values, e.g., yellow for 20 points or red for 10 points, do not matter and do not affect reading the figure. Therefore, for conciseness, we intentionally removed the color bars.

*7. Figure 6 indicates that the PDFs for NSMC radiance data largely deviate from a Gaussian distribution. Please discuss the potential reasons for this non-normal behavior in the text.*

Thank you for your careful review. Indeed, the asymmetrical distributions are commonly seen in gridded data. This asymmetry originates from misclassification of sky condition, e.g., due to parallax error, where a clear pixel is misidentified as cloudy (thus $x_P < y_H$). Additionally, the asymmetry may also be due to inaccurate input (e.g., aerosol, water vapor) used by the algorithm. We have added this discussion in the revised version.

*8. Figure 8(a) caption: Please specify which satellite sensor the observed reflectance data originates from.*

Thank you for your careful review. We have stated in the revised caption, "(a) Observed reflectance from the 0.65 μm visible channel of AGRI onboard FY-4B …"

*9. Table 8: Could you please verify the precision of the Mean Bias Error (MBE) values? Is the presented precision (two decimal places) justified given the accuracy of the underlying data?*

Thank you for your important remark; however, this does not concern our work. Indeed, all error tables shown in the paper limit the precision to integers, i.e., to 1 W m$^{-2}$, and for percentage error, to 1 decimal place. This level of precision is commonly employed in irradiance validation studies.

*10. At the end of this paper, the term "epistemological restraint" is unclear in this context. From a technical standpoint, the performance of FY-4B is more likely attributable to factors such as calibration quality and algorithmic limitations. It is suggested to reframe the argument using more conventional technical reasoning.*

Thank you for your suggestion, the term "epistemological restraint" is replaced by "care." The sentence is also rewritten.

*Reviewer #2:*

*This submission presents a unique radiation dataset in a monsoon-influenced continental climate. Then, using two sets of ground-based measurements and two sets of satellite irradiance retrievals, a thorough validation work is carried out. The idea is essential novel, since the uncertainty in the observations is rarely quantified, which makes this work unique. The Murphy-Winkler factorization is also used during validation, which is more comprehensive and more informative than the usual measure-based validation. The findings are interesting. I have only a few minor comments.*

Thank you for the positive remark. We have revised the work thoroughly according to your comments.

*1. In section 2.1.4, the algorithm used for Fengyun-4A is applied to Fengyun-4B to obtain the semi-empirical product. Is there any caution that one should take? For instance, does the calibration of sensors plays a part in this algorithm?*

Thank you for your comment. One of the advantages of semi-empirical retrieval methods is the independence of satellite calibration information. This has been noted in a recent work by Huang et al. (2025, Solar Energy, 296, 113593). We have added a statement in the revised version of the manuscript.

*2. The identity lines in some plots, e.g., Fig. 6, are hard to see. Perhaps the authors should consider adding some transparency to the color, such that the identity lines are visible.*

Thank you for your suggestion. The figures are revised accordingly.

*3. The paper can benefit from another round of proofreading, as some sentences are a bit long and hard to read.*

Thank you for your comment. We have carefully proofread the paper again, and the changes made are reflected in the revised version of the manuscript.

*4. The manuscript should minimize the use of semicolons, Such as in Lines 190-195, and elsewhere.*

Thank you for your comment. The semicolon usage is checked. Many of the semicolons were suggested by Grammarly, so we kept those.

*5. Line 135: the description of "CMA maintains more than 50,000 manned and unmanned weather stations" is not precise enough.*

Thank you for your comment. We have updated the number according to Reviewer#1's suggestion, i.e., 90,000 stations.

*6. The expectations in Eqs. (6) and (7) require more discussion, for example, how to compute them algorithmically?*

Thank you for your comment. The computation of the expectations is indeed not straightforward. We have, therefore, added an Appendix, elaborating the detailed computation procedure, see Appendix A in the revised version.

*7. Table 8 shows that combining two products reduces the error. Is this general? Or is it just for this particular location? More discussion on this is needed.*

Thank you for your careful review. Combining, as a strategy, has received much attention from the statistics community. Up to now, there is no complete theory defining when combination will work and when it will not. We have cited a couple of recent works on forecast (prediction) combination.

*8. This is a side topic: What is the station no. of QIQ? I am unable to find it on the BSRN website. Where can I access the data? Additionally, it would be good to include the geographical location of the site in Fig. 1 (this is optional, since the station is marked in Fig. 9).*

Thank you for this remark. The station no. of the QIQ station is 94. Since it is a new station, not all data are uploaded to the BSRN website. However, this should be available soon, per our recent communication with BSRN. For now, the data is hosted on Pangea, and the reviewer may access the data at:
https://doi.pangaea.de/10.1594/PANGAEA.979926

*Reviewer #3:*

*This manuscript presents a rigorous validation study comparing physical and statistical satellite irradiance retrieval methods against high- and low-accuracy ground observations. The work is timely, methodologically sound, and addresses a critical gap in solar energy meteorology by emphasizing the risks of using low-accuracy data for validation. The distribution-oriented validation framework is a notable strength, providing deeper insights beyond conventional metrics. The manuscript is well-structured, and the conclusions are supported by robust analysis. I recommend acceptance with minor revisions after addressing the following points.*

Thank you for reviewing our work and the confirmation on the contribution. We have revised the work thoroughly according to your comments.

*1. Fig. 2 is an important figure. However, in-text description is somewhat limited. The authors need to enhance the textual description of Fig. 2.*

Thank you for the remark. Indeed, the textual description of Fig. 2 is saved for brevity reasons. A detailed explanation can be found in Forstinger et al. (2021, Solar World Congress). We have cited the reference accordingly during the revision. However, if the reviewer finds it is important to discuss that fully here, we can add the text during the next iteration.

*2. Section 2.3.1: Besides MBE and RMSE, MAE is also another very popular metric. The authors should at least discuss the use MAE if not providing its results.*

Thank you for this interesting comments. Indeed, MAE is commonly used. However, as pointed out by Gneiting (2011, Journal of American Statistical Association, 106, 746-762), MAE and RMSE corresponds to different elicitable functional during verification. Therefore, if one is used, the other should not be used in general. This point is also echoed in numerous other works in solar energy, such as Mayer and Yang (2023, International Journal of Forecasting, 39, 981-991) or Yang and Kleissl (2023, International Journal of Forecasting, 39, 1640-1654). We have added some statements regarding this caveat in the revision.

*3. The capitalization of words in Eqs. 5, 6, and 7 should be consistent. Moreover, the expectations use in these equations should be further explained.*

Thank you for your careful review. The capitalization is now standardized. The expectation is more explained in Appendix A of the revised version (we added that to address a comment from another reviewer).

*4. Line 415-418: I cannot relate the in-text discussion to the results presented in Table 6. The authors should be more specific, i.e., from which numbers are various discussions made.*

Thank you for this suggestion. We have clarified the discussion as suggested.

*5. Eq. 9, the "-1" applied to (or the inverse of) a CDF results in a quantile function. This is not clearly stated, and should be added.*

Thank you for your comment. We have added a statement regarding the quantile function.

*6. The Heliosat-2 method is well known. The appendix seems redundant.*

Thank you for the remark. We agree with the reviewer that the working principle of Heliosat-2 is well known. However, the determination of upper and lower bounds of the dynamical range has various alternatives. In this work, we wish to explain our particular approach to determine the dynamical range, which makes the paper more self-contained. Therefore, the appendix is kept during the revision.